# Origami metamaterials for ultra-wideband and large-depth reflection modulation

Zicheng Song [1,2,8], Juan-Feng Zhu [3,8], Xianchao Wang[4,8], Ruicong Zhang[1,2], Pingping Min[1,2], Wenxin Cao[1,2], Yurong He[5], Jiecai Han[1], Tianyu Wang[5] ✉, Jiaqi Zhu [1,2] ✉, Lin Wu [3,6] ✉ & Cheng-Wei Qiu[7] ✉

The dynamic control of electromagnetic waves is a persistent pursuit in modern industrial development. The state-of-the-art dynamic devices suffer from limitations such as narrow bandwidth, limited modulation range, and expensive features. To address these issues, we fuse origami techniques with metamaterial design to achieve ultra-wideband and large-depth reflection modulation. Through a folding process, our proposed metamaterial achieves over 10-dB modulation depth over 4.96 – 38.8 GHz, with a fractional bandwidth of 155% and tolerance to incident angles and polarizations. Its ultra-wideband and large-depth reflection modulation performance is verified through experiments and analyzed through multipole decomposition theory. To enhance its practical applicability, transparent conductive films are introduced to the metamaterial, achieving high optical transparency (>87%) from visible to near-infrared light while maintaining cost-effectiveness. Benefiting from lightweight, foldability, and low-cost properties, our design shows promise for extensive satellite communication and optical window mobile communication management.

With the rapid advancement of the low-Earth orbit satellite internet technology, satellite communication has garnered increasing interest due to its affordable, high-speed, and low-latency internet access service. However, electromagnetic radiation and scattering from satellites can disrupt radio telescope observations of space, and the demanding space environment presents significant challenges in improving communication quality and managing satellite invisibility[1,2]. Although incorporating absorption into the mechanical structures has shown promise in reducing scattering fields and improving signal reception sensitivity, their opaqueness and the static nature of electromagnetic signal adjustments limit their applicability to diverse scenarios[3–7]. Hence, intelligent adaptive structures with dynamic control over

electromagnetic waves hold great promise for advancing satellite communication[8,9].

To achieve dynamic manipulation of electromagnetic waves, two main categories of methods have emerged: modulation of material electromagnetic parameters and changes in the structural shapes. The former involves introducing external stimuli, including the application of bias voltage[10–13], magnetic field[14–16], light irradiation[17–20], or heating[21–24], to intelligently control the electromagnetic properties of materials such as graphene, liquid crystal, and phase-transition material, allowing for the modulation. However, it is worth noting that despite the advantages, these devices often face bottlenecks in terms of narrow operation bandwidth and limited modulation depth[25–28]. Furthermore, these

¹Center for Composite Materials and Structures, Harbin Institute of Technology, Harbin 150080, China. ²Zhengzhou Research Institute, Harbin Institute of Technology, Zhengzhou 450018, China. ³Science, Mathematics and Technology, Singapore University of Technology and Design (SUTD), 487372 Singapore, Singapore. ⁴School of Mathematics, Harbin Institute of Technology, Harbin 150080, China. ⁵School of Energy Science & Engineering, Harbin Institute of Technology, Harbin 150080, China. ⁶Institute of High Performance Computing (IHPC), 138632 Singapore, Singapore. ⁷Department of Electrical and Computer Engineering, College of Design and Engineering, National University of Singapore, 117583 Singapore, Singapore. ⁸These authors contributed equally: Zicheng Song, Juan-Feng Zhu, Xianchao Wang. ✉e-mail: tianyu_wang@hit.edu.cn; zhujq@hit.edu.cn; lin_wu@sutd.edu.sg; chengwei.qiu@nus.edu.sg

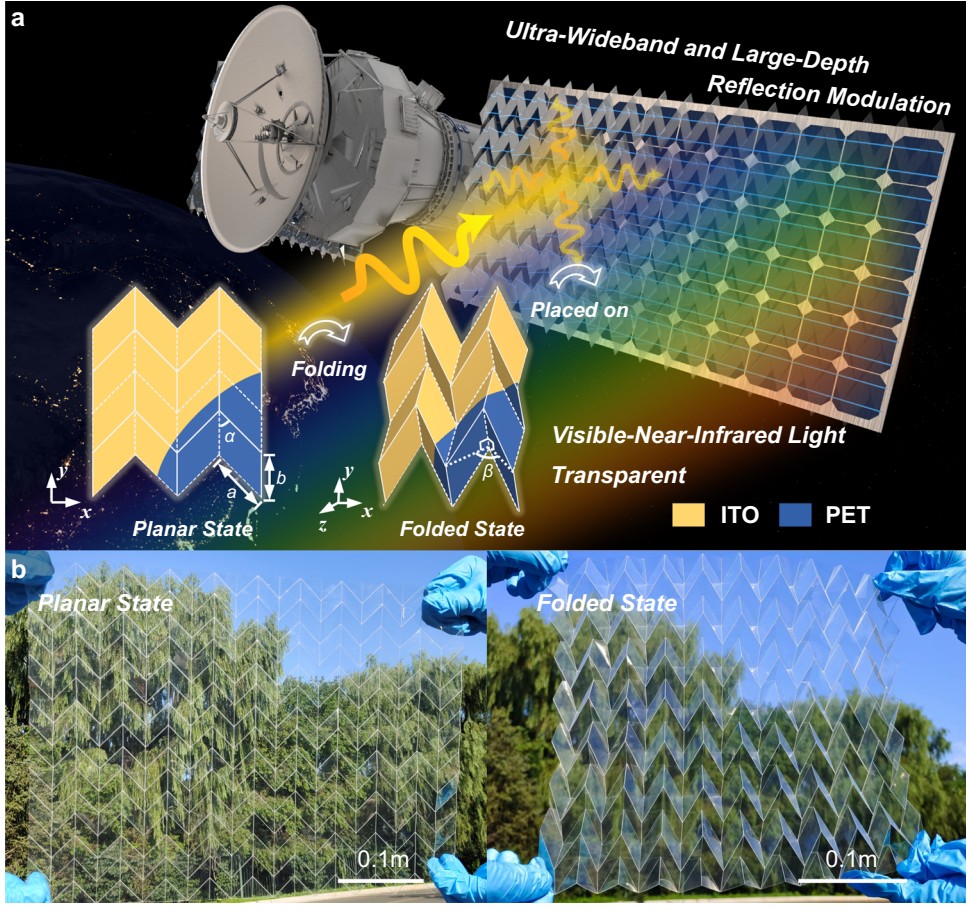

**Fig. 1 | Conceptual illustration and sample images of the proposed origami metamaterial. a** The scenario of proposed origami metamaterial configuration on a satellite's solar panel. **b** Images display the proposed origami metamaterial in its planar and folded states, respectively.

materials and delicate structures make these devices expensive and susceptible to external factors, such as temperature and vibration, which affect their applicability in the harsh space environment. Alternatively, the adjustable mechanical alteration of structures provides greater freedom regarding reconfigurable properties with robust characteristics. By applying an external mechanical force, the shape and size of the device can be arbitrarily controlled, thereby promising a higher degree of freedom in reconfigurability[29–31].

Origami structures are regarded as the most promising spaceborne devices among mechanical structures, owing to their distinctive transformable characteristics. Recently, the integration of origami with communication devices has yielded remarkable performance, including achievements such as light deflection[32], wideband switchable absorption (fractional bandwidth 90.1%)[33], and wide-angle absorption up to 70°[34]. These pioneering origami design efforts aim to improve reconfigurability and corresponding core performances without taking optical transparency and process complexity into account[35–37]. However, the practical spaceborne application of origami metamaterials has encountered challenges, especially in achieving optical transparency, ultra-wideband, and large-depth reflection modulation simultaneously.

In this work, we employ origami techniques and flexible electronic materials in metamaterial design to achieve large-depth reflection modulation over an ultra-wideband. By applying an external stimulus, we induce a Miura-fold transition in a planar sheet, resulting in a transition from strong reflection near 0 dB to weak reflection <−10 dB within the 4.96 – 38.8 GHz frequency range, leading to 10-dB modulation depth and 155% fractional bandwidth. Capitalizing on transparent conductive films, proposed metamaterial enabling over 87.2% transmittance covering visible to near-infrared range. To gain insight

into the mechanism, a multipole decomposition theory is employed to analyze the ultra-wideband and large-depth modulation behavior of the origami metamaterial. Using laser etching crease on indium tin oxide (ITO) – polyethylene terephthalate (PET) sheet, we fabricate the origami metamaterial and verify its transparency, ultra-wideband, and large-depth reflection modulation through actual experiment. In this work, the proposed origami metamaterial cost is just 16 $/m², thanks to excluding precious metals in the fabrication process. Further, the proposed metamaterial is slightly heavier than an A4 paper of the same size, making it a lightweight and low-cost solution for space communication and optical window mobile communication management applications.

## Results

### Design and verification of origami metamaterial

Figure 1a depicts the proposed origami metamaterial structure. In the cross-sectional view of the $2 \times 2$ unit cells, the solid (dashed) line represents a mountain (valley) crease, which enables the conversion of the planar sheet to the Miura-fold structure. Due to the robust mechanical performance under large-temperature differences, the origami substrate uses 0.125 mm-thick PET and is covered by a 20 nm-thick ITO film with a surface resistance of 150 Ω/sq. Four parallelogram resonators of a unit cell are divided by creases. The parameters of the parallelogram resonators are fixed as $a = 25 \times \sqrt{2}$ mm, $b = 25$ mm, and parallelogram angle $\alpha = 45°$. During the folding process, we assume that the Miura-fold structure is made of an ideal material with an infinite tensile modulus, ensuring that each parallelogram maintains a rigid plane. The single-degree-of-freedom nature of the Miura-fold structure allows its shape to be uniquely determined by the folded

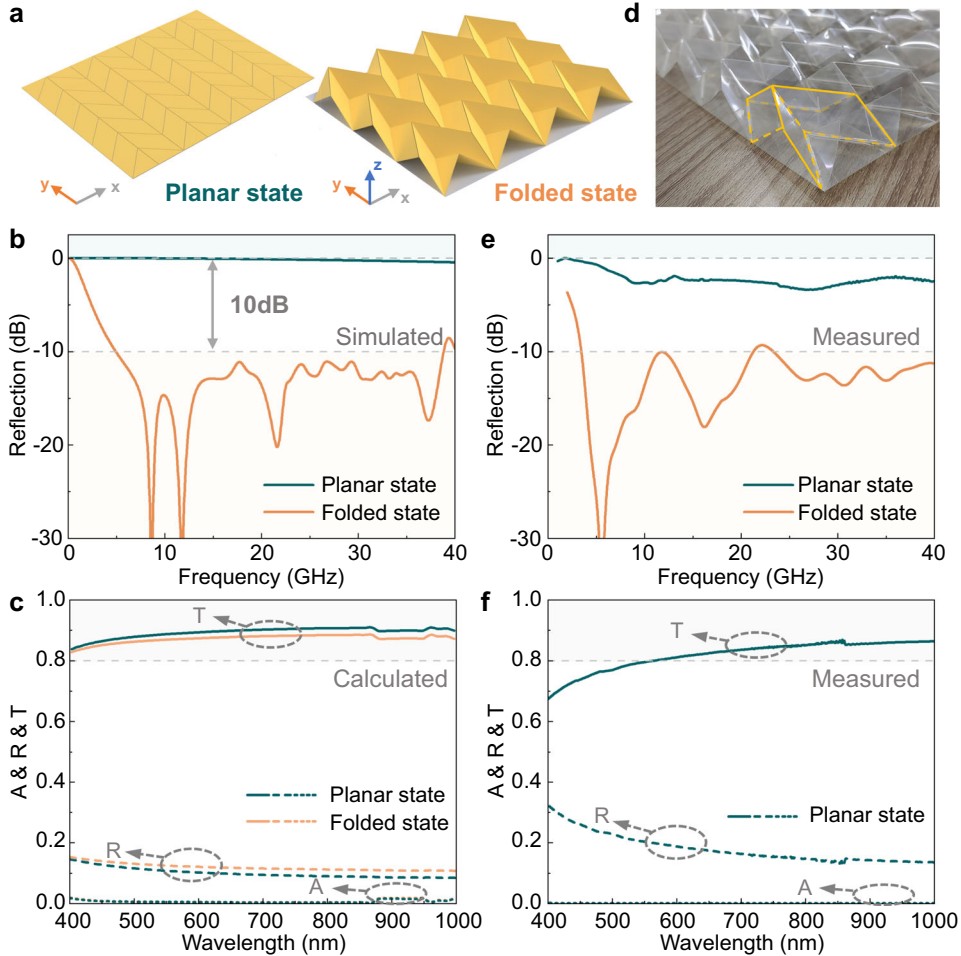

**Fig. 2 | The simulated and measured spectrum over microwave, visible and infrared light of proposed origami metamaterial in its planar and folded states. a** Schematic of the origami metamaterial in its planar and folded state. **b, e** The simulated and measured microwave reflection of proposed metamaterial in its planar and folded state. **c, f** The proposed metamaterial's calculated and measured absorptance (A), reflectance (R), and transmittance (T) in visible to near-infrared light. **d** The image of the fabricated metamaterial's unit cell in its folded state.

angle $\beta$, which is the dihedral angle between two adjacent parallelograms. By using the genetic algorithm, the reflection bandwidth below −10 dB is maximized for wideband absorption while also maximizing the bandwidth of reflections above −1 dB to achieve wideband reflection[38]. Hence, the folded state and planar state of origami metamaterial are determined to be $\beta = 95°$ and 180°, respectively. The primary mechanical structure is proposed for adjusting the working states of origami metamaterials, which is described in Supplementary Information S1. The image of the proposed optical transparent origami metamaterial in its planar and folded state is shown in Fig. 1b.

Using the commercial software, we solve Maxwell's equations to obtain the microwave reflection spectra of the proposed metamaterial in its different state. The configuration of the metamaterial at planar and folded state is illustrated in Fig. 2a. The periodic boundary is adopted in both x- and y- directions to simulate the infinite array effect. Since the solar panels have a high conducting coating that collects the generated current, it can be considered as an electric boundary at the bottom of the origami metamaterial. Hence, the reflection of the metamaterial can be obtained when the y-polarized plane wave is normally incident. Adopting the electric boundary allows the neglect of transmittance; thus, absorption and reflection are equal to total power. As illustrated in Fig. 2b, the proposed metamaterial achieves strong reflection near 0 dB in the 0–40 GHz range with a fractional bandwidth of 200% in its planar state. In the folded state, it achieves ultra-wideband weak reflection of less than −10 dB in the 4.96 –

38.8 GHz band, with a fractional bandwidth of 155%. Hence, the proposed structure achieves over 10-dB-depth reflection modulation over 155% ultra-wideband through the folding process.

Due to the large electric size of the structure in the visible-near-infrared light range, performing full-wave simulations becomes impractical. Instead, the matrix theory is used to calculate the structure's absorptance, reflectance, and transmittance in its planar and folded states under visible-near-infrared light normal incidence. The measured complex refractive indices of ITO and PET[39,40] are used to construct the wave-transfer matrix and then obtain the transmittance and reflectance of the structure through the scattering matrix, detailed in Supplementary Information S2. Figure 2c presents the unpolarized optical spectrums of metamaterial in its planar and folded states. The optical transmittance slightly decreases when the structure is transformed from the planar to the folded state, with the average transmittance in the folded state and planar state being 87.2 % and 88.6 %, respectively. The slight attenuation of transmittance is mainly due to the reflectance, while absorptance contributes little.

To validate the design, we fabricate the origami metamaterial sample using 400 mm × 500 mm ITO-PET sheets with a surface resistance of 153.5 Ω/sq, resulting in an 8×10 array configuration. The manufactured metamaterial and its detailed unit cell in its folded state are shown in Fig. 1b and Fig. 2d, respectively. The measurement result presented in Fig. 2e reveals that the proposed metamaterial demonstrates a reflection of over −3 dB (with an average of −2.47 dB) in its

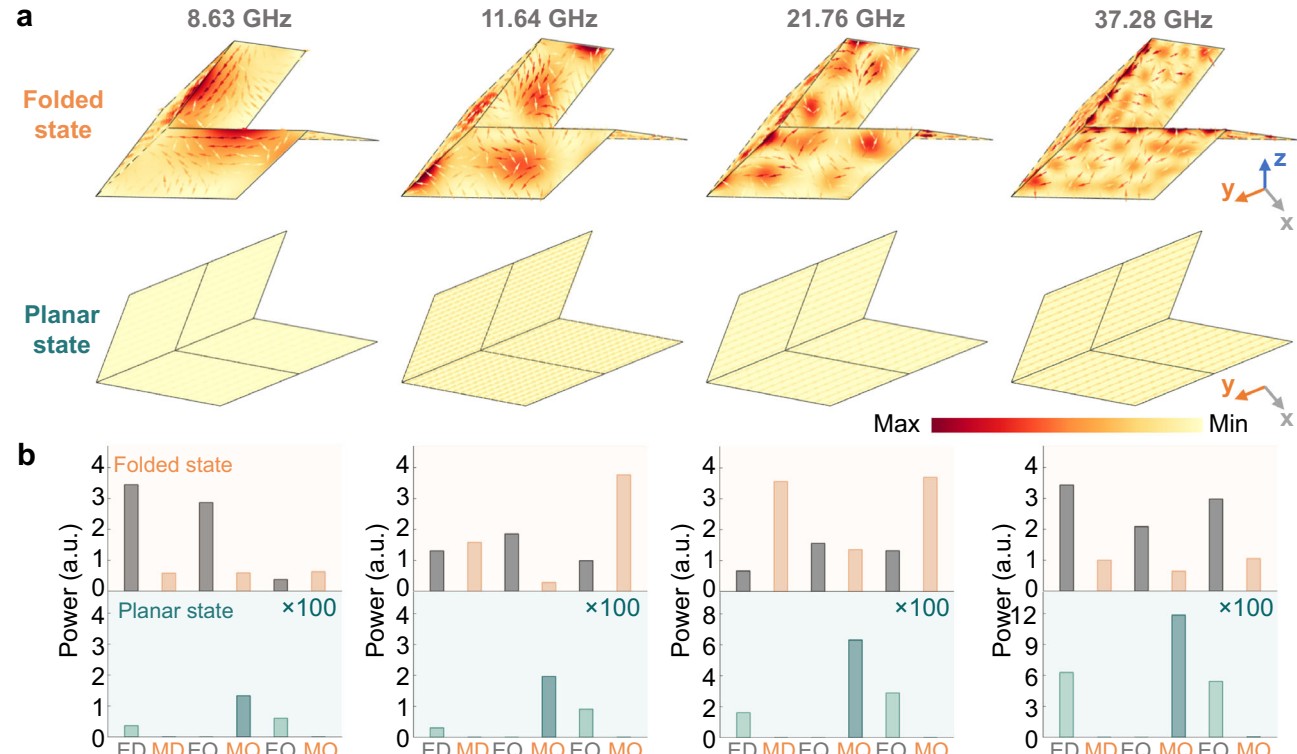

**Fig. 3 | Ultra-wideband and large-depth reflection modulation analysis of proposed metamaterial in its folded and planar state. a** Distribution of the surface current (arrows) and the surface power loss density (background color) of the proposed metamaterial unit cell in planar and folded state at 8.63, 11.64, 21.76, and 37.28 GHz. **b** The multipole decomposition results of the proposed metamaterial unit in folded/planar states at 8.63, 11.64, 21.76, and 37.28 GHz.

planar state and a reflection of less than −10 dB (with an average of −14 dB) in its folded state, resulting in an average modulation depth of 11.53 dB. The disparities observed between the simulation and measurement results can be attributed to the mechanical errors inherent in the structure, as its unit cells are not guaranteed to be rigid planars due to the utilization of actual materials. Nevertheless, despite some discrepancies, the preliminary verification highlights the significant potential of origami metamaterials in achieving ultra-wideband and large-depth reflection modulation capabilities.

The proposed structure's visible-near-infrared light reflectance and transmittance are measured using a spectrophotometer to assess the transparency. Figure 2f shows that the proposed structure achieves high transmittance at wavelengths between 400–1000 nm, with an average transmittance exceeding 81.8%. The transmittance attenuates mainly due to the high reflection among the visible to infrared light, and the absorption is negligible. Subsequently, the transparency of the origami metamaterial in both planar and folded states is verified through a simulated sunlight experiment, with the detailed configuration provided in Supplementary Information S3. The solar panel achieves an energy supply of 9.90 mW without shade under simulated sunlight irradiation. When shaded by the origami metamaterial in both its folded and planar states, the solar panel achieves outputs of 8.92 mW and 8.70 mW, corresponding to efficiencies of 90.1% and 87.9%, respectively. Simulated sunlight experiment results indicate that the presence of origami metamaterial does not significantly affect the energy generation of the solar panel. With the robust high transmittance in visible-near infrared light, the proposed origami metamaterial is promising in broader applications requiring optical transparency.

### Principle analysis of ultra-wideband and large-depth reflection modulation

To clarify the principle of ultra-wideband and large-depth reflection modulation of origami metamaterial, Fig. 3a shows the unit cell's surface loss density and surface current, corresponding to four reflection dips at 8.63, 11.64, 21.76, and 37.28 GHz. At the lower resonant frequency, the folded state exhibits high loss due to the surface current flows along the upper V-shape edge. In contrast, the planar state shows minimal surface currents. As the frequency increases, the folded state exhibits a complex loss density distribution resembling a soda-biscuit shape, while the planar state maintains low loss. The results show that the proposed metamaterial facilitates strong surface currents in its folded state, resulting in high absorption across a wideband frequency range. When the metamaterial deforms to its planar state, the surface current and absorption weaken, resulting in a significant increase in reflection across a wideband frequency range.

The theory of multipole decomposition further provides valuable physics insights into the phenomenon of the proposed origami metamaterial. By solving the current density of the origami structure as input through commercial software, multipole decomposition theory is used to classify the currents and determine their contributions to the total power, as illustrated in Eq. (1).

$$P_{\text{total}} = P_{ED} + P_{MD} + P_{EQ} + P_{MQ} + P_{EO} + P_{MO} \tag{1}$$

where $P_{ED}$ and $P_{MD}$ represent the power of electric and magnetic dipole, $P_{EQ}$ and $P_{MQ}$ represent the power of electric and magnetic quadrupole, $P_{EO}$ and $P_{MO}$ represent electric and magnetic octupole, respectively. Further detailed derivation is provided in Supplementary Information S4. Figure 3b illustrates the multipole decomposition results for the planar and folded unit cell structures. Among these resonances, the power of modes in the folded state is significantly higher than that of the planar state, indicating the efficient excitation of modes in the folded structure under plane wave irradiation.

The comparison of mode powers offers valuable insight into the proposed metamaterial's ultra-wideband and large-depth reflection modulation phenomenon. At the resonant frequency of 8.63 GHz, the

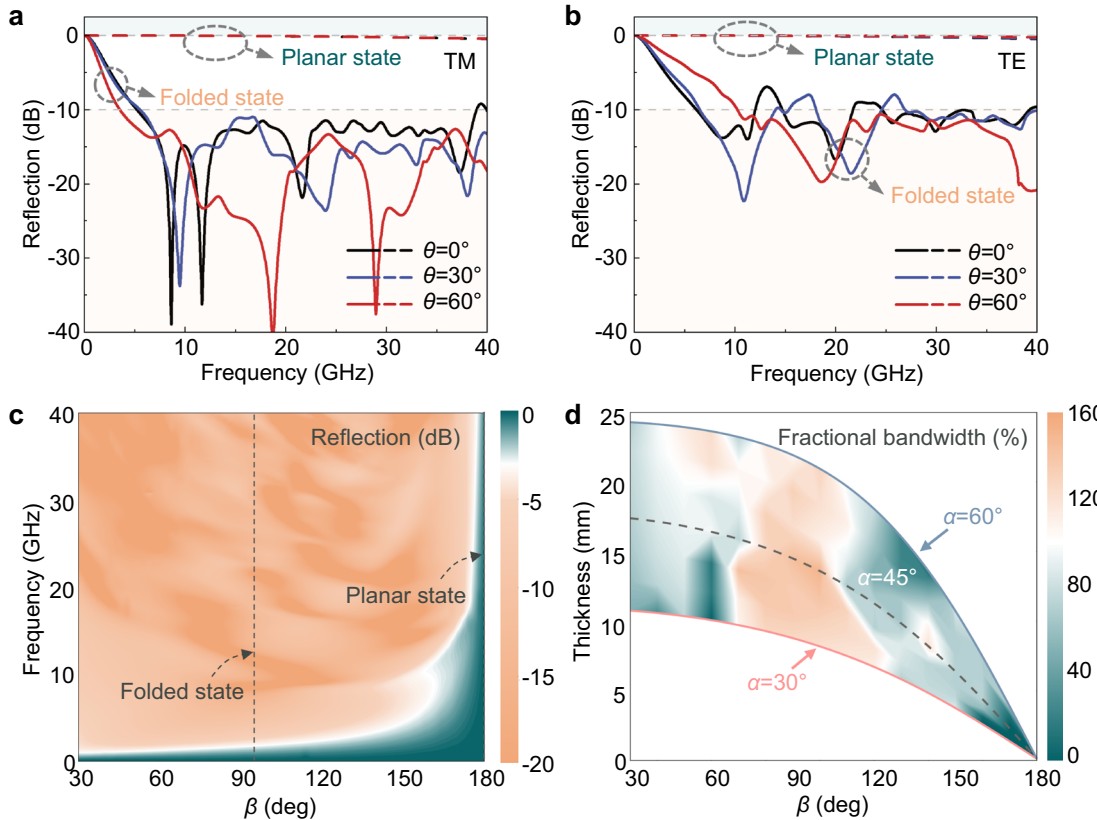

**Fig. 4 | The reflection response of proposed origami metamaterial varies with different folded angles and oblique angles. a** The reflection spectral of proposed metamaterial under TM wave oblique incidence. **b** The reflection spectral of proposed metamaterial under TE wave oblique incidence. **c** The reflection spectral of proposed metamaterial with different folded angle $\beta$. **d** The reflection lower than −10dB bandwidth of proposed metamaterial as a function of the folded angle $\beta$ and the parallelogram angle $\alpha$.

folded state unit cell's dominant modes are electric dipole and quadrupole. While electric dipole, quadrupole, and octupole modes contribute significantly at 37.28 GHz, indicating the prevalence of electric modes at these resonances. At 11.64 GHz, magnetic octupole modes play a significant role in absorption, accompanied by the contribution of electric modes. These combined factors indicate the collaborative influence of both electric and magnetic modes on the resonance. The magnetic dipole and octupoles exhibit high mode power, indicating the magnetic modes dominate at 21.76 GHz. Four typical resonant frequencies are analyzed emphatically, and similar strong mode behaviors are observed across a wide range of frequencies beyond the ones highlighted, which will not be shown in detail for brevity. Hence, robust electric and magnetic mode resonances over a continuous frequency band in the folded state lead to ultra-wideband absorption. Conversely, the emergence of weak mode resonances in the planar state results in ultra-wideband reflection. With the aid of multipole decomposition, the complex field and current distribution of proposed origami metamaterial are precise analyses, revealing the ultra-wideband and large-depth reflection modulation behavior.

## Characterization and analysis of practical applicability

To assess the practical applicability of the proposed metamaterial, we further study the electromagnetic reflection of the structure under oblique incidence conditions. Figure 4a illustrates the widening of the bandwidth of the proposed metamaterial in its folded state as the TM plane wave incidence angle $\theta$ increases. At a 30° oblique incidence, the peak near 38 GHz disappears, while at a 60° oblique incidence, the reflection band less than −10 dB further shifts towards lower frequencies, resulting in ultra-wideband weak-reflection ranging from 3.56 to 40 GHz, with 167% fractional bandwidth. Furthermore, even at

up to 60° oblique incidence, the reflection of the proposed metamaterial at the planar state remains near 0 dB. Figure 4b shows that the structure exhibits similar stable spectral characteristics near 0 dB under cross-polarization oblique incidence in its planar state. Although minor peaks can be observed in the reflection spectra in the folded state, the reflection in most frequency bands is still below −10 dB. Hence, the proposed origami metamaterial is suitable for achieving ultra-wideband and large-depth reflection modulation under oblique incidence. These findings highlight the proposed origami metamaterial's capacity to maintain sufficient oblique incidence and polarization stability, thus demonstrating its practical applicability.

Figure 4c presents the reflection spectra of the proposed metamaterial as the folded angle $\beta$ changes. The reflection shows slight variation as $\beta$ ranges from 30° to 130°, indicating sufficient mechanical tolerance for achieving wideband weak reflection in its folded state. As $\beta$ increases beyond 130°, the weak-reflection frequency band shifts towards higher frequencies while the reflection gradually enhances and reaches its maximum at 180°. To explore the influence of geometrical parameters on the wideband weak-reflection performance of origami metamaterial, we analyze the fractional bandwidth of reflection less than −10 dB as a function of parallelogram angle $\alpha$, thickness, and folded angle $\beta$, as depicted in Fig. 4d. For the proposed origami structure ($\alpha = 45°$), the fractional bandwidth exceeds 100% within the range of folded angle $\beta$ between 80 and 110°. The maximum fractional bandwidth of 155% is achieved at $\beta = 95°$. The ultra-wideband weak-reflection characteristic of the metamaterial in its folded state arises from the absorption, thus adhering to Rozanov's limitation. This limitation constrains the absorption bandwidth ratio to the structure's thickness, implying that thinner structures tend to have narrower absorption bandwidths[41]. When the folded angle $\beta$ is between 80 and

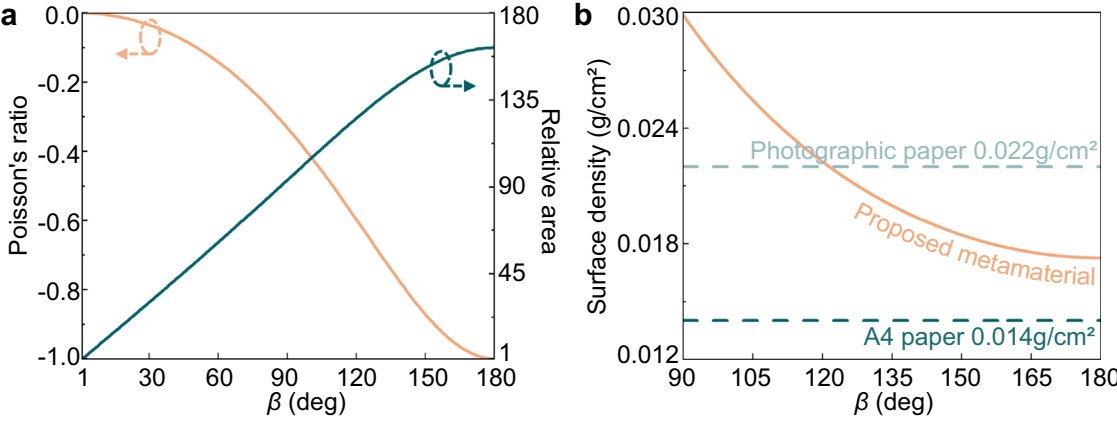

**Fig. 5 | Mechanical properties of proposed origami metamaterial. a** Calculated Poisson's ratio and relative area as a function of different folded angle $\beta$. **b** The comparison of the proposed metamaterial's surface density with different folded angle $\beta$ and conventional paper surface density.

## Table 1 | Comparison of different approaches to dynamic modulation in metamaterials

| Ref. | Approach | Material | Operation frequency | Modulation depth [a] | Transparency | Cost ($/m²) |
|------|----------|----------|---------------------|----------------------|--------------|-------------|
| 43 | Carrier doping | Graphene | 7 – 18 GHz (88%) | Over 3 dB | Yes | 96,950 [b] |
| 26 | | Graphene | 10 GHz | 50 dB | Yes | 96,950 [b] |
| 44 | Phase transition | VO$_2$ | 9.36,18.6 GHz | 11.15 dB | Yes | Not report |
| 45 | | Liquid crystal | 2.62 THz | Over 3 dB | Yes | > 348 [c] |
| 33 | Mechanics | Resistance ink | 6 – 16 GHz (90.9%) | Over 10 dB | No | Not report |
| **Ours** | | **Indium tin oxide** | **4.96 – 38.8 GHz (155%)** | **Over 10 dB** | **Yes (> 87%)** | **16** |

[a] The modulation depth is defined as $P_{max} - P_{min}$, where $P_{max}$ and $P_{min}$ are the maximum and minimum of reflection in the dB scale.

[b] The graphene structures consist of bilayer graphene, single layer graphene price is $ 48475/m², which can be found at www.graphene-supermarket.com.

[c] The price of liquid crystal 4′-n-pentyl-4-cyanobiphenyl (5CB) $174/g, which can be found at www.sigmaaldrich.com. Estimated use of 2 g of liquid crystal per square meter of structure, while the substrates and fabrication cost are not considered.

110°, the structure's thickness remains relatively unchanged, providing favorable conditions for wideband absorption. However, as $\beta$ further increases beyond this range, the structure's thickness decreases significantly, reducing absorption bandwidth. When $\beta$ reaches 180°, absorption bandwidth and thickness reach the minimum value, leading to a strong reflection. The trend of reflection bandwidth is similar in origami metamaterials with parallelogram angles $\alpha$ ranging from 30° to 60°, confirming the robust ultra-wideband and large-depth reflection modulation.

While the absorption varies with thickness, the proposed origami metamaterial significantly distinguishes itself from classic Salisbury screens. As an example, the Salisbury screen with origami mechanical support exhibits intermittent narrowband absorption, which depends on interlayer interference, detailed in Supplementary Information S5. As clarified by multipole decomposition, the origami metamaterial in its folded state can effectively achieve ultra-wideband absorption due to continuous mode resonances.

Figure 5a illustrates the foldability of the proposed metamaterial. The Poisson's ratio of the structure is defined as $v = -(dl/l)/(dw/w)$, where $l$ and $w$ represent the length and width of the origami metamaterial unit cell, respectively. It can be observed that as the $\beta$ increases, the Poisson's ratio of the structure gradually changes from 0 to −1. A negative Poisson's ratio indicates that when the structure expands in one direction, it also expands in the perpendicular direction. This property is advantageous for metamaterial adapting to cover large-sized solar panels within limited storage space. The relative area parameter offers a more intuitive representation of the change in size. By comparing the reference area at a 1° folded angle, the structure gradually expands and reaches a maximum of 162 times in its planar

state. The actual and simulated unfolding processes are shown in Supplementary Movie 1 and 2, respectively. Meanwhile, we compare the surface density of the proposed metamaterial with that of commonly used A4 paper and photographic paper in Fig. 5b. In the folded state with a folded angle $\beta = 95°$, the structure is slightly heavier than photographic paper. In contrast, in the planar state, the surface density of the structure falls between that of photographic paper and A4 paper. The structure's exceptionally lightweight nature and foldability make it highly suitable for satellite applications that demand limited storage and expandable capabilities.

Most importantly, the proposed metamaterial is adopted affordable flexible electronic materials ITO-PET sheet. Due to no precious metal being used, its material cost is significantly reduced. Additionally, using a laser for etching creases further reduces the fabrication cost and shortens the processing time by avoiding the need to etch customized resonators on the surface. Compared with the 96,950 $/m² cost for graphene dynamic metamaterial, the sample used in this work costs only 1.65/10,000 of it, equivalent to 16 $/m². This price is anticipated to be further reduced through process improvement. As demonstrated in Table 1, the proposed origami metamaterial offers significant improvements in operational bandwidth and modulation depth compared to previous dynamic metamaterials. Simultaneously, the proposed origami metamaterial possesses advantages such as high transparency, lightweight, and foldability, providing a practical path for spaceborne dynamic modulation applications.

## Discussion

This work presents an origami metamaterial demonstrating ultra-wideband and large-depth reflection modulation while maintaining

high transparency to visible light and near-infrared wavelengths. By employing an external stimulus to switch between folded and planar state, the proposed metamaterial alters its reflection characteristics from less than −10 dB to close to 0 dB within the frequency range of 4.96−38.8 GHz. As a result, it leads to an ultra-wideband performance with a bandwidth of 155% and a large modulation depth of over 10 dB. Furthermore, a transparent conductive film enables the proposed structure to maintain an average transmittance of over 87.2% in both states, encompassing the visible to near-infrared range. The reflection modulation and transparency are verified through experiments, demonstrating its potential for practical applications. To gain further insight into the mechanism behind the ultra-wideband modulation behavior, we introduce the theory of multipole decomposition, which provides a precise mode contribution analysis of the origami meta-material. Moreover, the proposed structure exhibits excellent stability under different polarization and oblique incidence while demon-strating foldability and lightweight performance. In addition to remarkable performances, the proposed metamaterial only costs 16 \$/m² , much cheaper than other dynamic metamaterials. The proposed origami metamaterial provides a practical path for applications in space communication and optical window mobile communication management.

Behind the robust performance and cost-effectiveness of origami metamaterials lies the boundless potential of multidisciplinary inte-gration. It is noted that typical metamaterial functions, such as polar-ization conversion and beam deflection, can be seamlessly fused with origami technology. The exciting combination promises to introduce advanced dynamic modulation characteristics, which can serve as a novel design dimension and offer innovative solutions for space-borne communication devices and avoiding interference with radio tele-scopes. Furthermore, considering the origami metamaterial as a design platform, the exciting integration with flexible electronic materials will likely unlock new possibilities and drive advancements in dynamic modulation for devices requiring transparency and various wearable electronic devices.

## Methods
### Numerical simulations
Full wave simulations are performed using the commercial software of CST Microwave Studio 2020. In all simulations, the unit cell boundary is used in both the x- and y- directions; the electric boundary is adopted at the -z direction, while the open boundary is adopted at the +z direction. The PET substrate has a dielectric constant of $\varepsilon_r = 2.65(1 - j0.005)$[42], while the 20 nm-thick ITO film has a surface resistance of 150 Ω/sq.

### Origami metamaterial fabrication
The origami metamaterial is fabricated through the micro-precision laser etching equipment from Yuanlu Optoelectronics. Specifically, a single infrared (1064 nm) nanosecond pulse laser is employed to etch blind slots on a 400 mm × 500 mm ITO-PET sheet with a surface resistance of 153.5 Ω/sq. These blind slots are etched to create folding creases with a width of 0.5 mm and a depth of 0.05 mm. Subsequently, the sheet is manually folded along the folding creases, creating origami metamaterial in the folded state. During the microwave measurement, the origami metamaterial is fastened to a 0.1 mm-thick PET sheet through wires to maintain the intended folded angle $\beta = 95°$.

### Microwave reflection measurement
To verify the reflection modulation performance of the origami metamaterial at different states, we use three pairs of wideband horn antennas (1−18, 18−27, 27−40 GHz) connecting vector network analy-zer. For normalization purposes, two metal plates of the same size as the planar and folded states were used.

### Visible−near−infrared reflectance and transmittance measurement
The optical spectrum measurement of the origami metamaterial is conducted using the Perkin Elmer Lambda 750 S, operating under the wavelength range of 400−1000 nm. With the assistance of a 60 mm integrating sphere, transmittance and reflectance are simultaneously measured. Due to the limitation of the test fixture, the optical spec-trum of a 1.5 × 1.5 cm² square planar state sample at normal incidence is measured as a representative measurement instead of the entire sample.

### Simulated sunlight experiment
To evaluate the transparency of the origami metamaterial, a Merry Change solar simulator MC-SSA100 is used for measuring the power of a solar panel with or without metamaterial influence. Covering the sunlight spectrum, a light source equivalent to triple solar intensity (300 mW/cm²) illuminates the solar panel's 12 × 12cm² area vertically. By using a multimeter to measure the open circuit voltage and short circuit current of the solar panel under light radiation to obtain the maximum output power.

## Data availability
All data supporting the findings of this study are available within the article and its supplementary files. Any additional requests for infor-mation can be directed to, and will be fulfilled by, the lead contact. Source data are provided with this paper.

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

## Acknowledgements

This work was supported by the China National Key R&D Program during the 14th Five-year Plan Period (Grant No. 2023YFB3811600); Major Program of Harbin Institute of Technology (Grant NO. 2023FRFK01002); Key Project of National Natural Science Foundation of China (Grant No. 52032004); National Natural Science Foundation for Distinguished Young Scholars of China (Grant No.51625201); National Youth Science Funds of China (Grant No. 52102039); Key Research and Development Program of Heilongjiang Province (Grant No. GA21D001); Fundamental Research Funds for the Central Universities (Grant No. HIT.O-CEF.2022011); Fundamental Research Funds for the Central Universities (Grant No. 2022FRFK060026). Wu gratefully acknowledges the Start-Up Research Grant (Grant No. SRG SMT 2021 169) and Kickstarter Initiative (SKI) (Grant no. SKI 2021_02_14) from the Singapore University of Technology and Design; and National Research Foundation Singapore (Grant No. NRF2021-QEP2-02-P03, NRF2021-QEP2-03-P09, and NRF-CRP26-2021-0004). Qiu gratefully acknowledges the MTC Programmatic Fund (Grant No. A-8001285-00-00). Song also wants to thank the China Scholarship Council under grant no. 202206120100, studying at the Singapore University of Technology and Design.

## Author contributions

Z.S. conceived and planned the research. Z.S., J.Z. and X.W. conducted the theoretical studies and simulations. Z.S., R.Z. and P.M. fabricated the samples and performed the experiments, Z.S., W.C., Y.H., J.H., T.W., J.Z., L.W. and C.W.Q. discussed the results. Z.S. wrote the paper with input from all the authors.

## Competing interests

The authors declare no competing interests.
