## [Peer Review File · Nature Communications]

Origami Metamaterials for Ultra-Wideband and Large-Depth Reflection ModulationREVIEWER COMMENTS

Reviewer #1 (Remarks to the Author):

< Origami Metamaterials for Ultra-Wideband and Large-Depth Reflection Modulation >

In this work, the authors propose an origami metamaterial utilizing Miura-ori units and fabricated from ITO-PET sheets. The proposed metamaterial achieves a modulation depth of over 10 dB and a fractional bandwidth of 155% within the frequency range of 4.96 GHz to 38.8 GHz, exhibiting tolerance to incident angles and polarization. In the planar state, it achieves strong reflection close to 0 dB over the range of 0-40 GHz, with a fractional bandwidth of 200%. In the folded state, it achieves weak ultra-wideband reflection of less than -10 dB within the frequency range of 4.96-38.8 GHz, with a fractional bandwidth of 155%. Capitalizing on transparent conductive films, high transmittance in the wavelength range of 400-1000 nm is achieved. The authors experimentally verify the ultra-wideband and large-depth reflection modulation performance and analyze it using the theory of multipole decomposition.

There are some issues need to be addressed before it can be further considered.

1. What is the indispensable role of origami in this absorber design? Does the deformation only provide a change in thickness (or the distance to the metallic background), thus improve the absorption bandwidth? If so, this mechanism seems identical to conventional Salisbury screen or Jauman absorbers. The spacer thickness can be easily controlled by filling with foam or dielectric slabs, which has been extensively studied before.
2. If Miura-ori only provides thickness as mechanical support, with a uniform impedance film placed on top, what are the differences in the absorption performance compared to the device described in the paper?
3. Is the metal plate transparent in the wavelength range of 400-1000 nm? In my understanding, the metal background layer is crucial to the absorption performance. Its light transmittance must be discussed if the advantage of optical transparency is claimed.
4. How does the metal background deform together with the origami layer?
5. The Poisson's ratio and Relative area in Figure 5 have both been well studied. What makes them particularly noteworthy or special?

Overall comment: major revision

Reviewer #2 (Remarks to the Author):

The authors reported an origami ITO-PET metamaterial with a broadband and large depth reflection modulation function. It was found that when inducing a Miura-fold transition in the metamaterial, a transition from strong reflection of near 0 dB to weak reflection of less than -10 dB could be reached within the frequency range from 4.96 to 38.8 GHz. Moreover, high transmittance of over 87.2% was reached for the metamaterial both in planar and folded states within the wavelength range from visible to near-infrared wavelength.

These results were interesting, and the evidences were solid. The manuscript could be accepted after minor revision.

1. How about the application scenario of the proposed origami ITO-PET metamaterial in satellite communication. This should be carefully discussed in the manuscript.

Reviewer #3 (Remarks to the Author):

The authors sputtered conductive materials on the surface of the Mirua-ori structure, achieved microwave band modulation by changing the folding angle, and chose transparent material to achieve visible and infrared band transmission. I recommend rejecting the article for publication in Nature Communication for the following reasons.

- (1)The idea of sputtering conductive materials on origami structures for wave-absorbing modulation was not innovative enough, although transparent wave-absorbing materials were chosen.
- (2)The origami wave-absorbing structure in the paper has a reflectivity -10 dB bandwidth that does not compare favourably with other wave-absorbing structures at the same thickness, and is sensitive to multi-angle incidence and polarisation of plane waves.
- (3)Miura-ori is negative Poisson's ratio structure, which cannot completely cover the target metal surface during the folding modulation process.
- (4)There is no specific solution for the control of the origami structure; the PET crease is plastic and it may not be easy to maintain the exact folding angle when controlling the whole structure.
- (5)The design of this wave-absorbing origami structure does not take into account its reliability, the difference in temperature between day and night in space applications can be up to 300°C, which PET may not be able to cope with, as well as the effects of high-energy rays from space.

Reviewer #4 (Remarks to the Author):

The work leverages origami structure to create a foldable metamaterial that enables electromagnetic ultra-wideband and deep reflection modulation. The optimized material can achieve a modulation depth exceeding 10 dB within the microwave telecommunication range, showcasing its potential for cost-effective and lightweight application in satellite frequency bands. The findings and methodology are sound. Given that various origami structures have previously been employed in the creation of reconfigurable electromagnetic devices with diverse functions, particularly in microwave range [Adv. Mater.2017, 29, 1700412; J. Phys. D: Appl. Phys. 54, 165111 (2021); Opt. Lett. 46, 1349 (2021); IEEE Trans. Antennas Propagat. 70, 4558–4568 (2022); Sci. Rep. 12, 13449 (2022)]. The work shows promise in practical application, but it needs to differentiate itself from prior work in the field. Therefore, I am uncertain if the significance of this work justifies its publication in Nature Communications.

The following comments are listed for the authors' reference.

- The authors referred to the optimal beta angle for achieving wideband weak or wideband strong reflection through the optimization process on Page 4. Could the authors provide more details about the specific optimization technique employed in this process?
- This work emphasizes its application in microwave telecommunication. Considering that the metamaterial's 'floor area' is reduced when folded, how can we ensure a fair and accurate evaluation of the material's performance across different folded angles, for example, energy output?
- Referring to both Fig. 2b and Fig. 2c, it's evident that the spectral features in the microwave range and the near-infrared range are notably distinct. I'm curious about how the reflectance spectrum might differ if matrix theory were utilized instead of full-wave simulations to investigate these features in the microwave range.
- Given that both the planar and folded states demonstrate high transmittance in the visible and near-infrared range (as shown in Fig. 2c), what specific advantages does the origami metamaterial offer in this context?
- The color variation in the colorbar, ranging from -3 to -20 dB in Figure 4c, appears limited. Enhancing the colorbar settings could potentially provide a richer representation of the data.

Additionally, in Figure 4d, there seem to be abrupt transitions in the colormap. It may be beneficial to consider refining the plot or providing additional explanation for these sharp changes.

- While the reported structure can be expanded in one direction, it would be interesting to know if the folded structure, which can be expanded along both the x- and y-directions, exhibits similar properties. Have these properties been investigated or considered?
- It would be helpful to include the full names of the labels 'A,' 'R,' and 'T' in the caption of Figure 2 or within the main text for clarity and better understanding.
- Since Equation 1 is featured in the main text, it would be beneficial to provide the definition of each component within the main text, even though they are already given in the supplementary materials. This would enhance the accessibility of the information for readers.
- Consider revising certain terms or sentences to enhance accessibility for general readers. For example, "non-pol", "Visible-near-infrared light measurement".
- Gramme errors. For example, "In the range of folded angle beta is between 80 and 110°", "Comparison of different approach to dynamic modulation metamaterial".

Replies to the Reviewer's Comments

Reviewer #1:

In this work, the authors propose an origami metamaterial utilizing Miura-ori units and fabricated from ITO-PET sheets. The proposed metamaterial achieves a modulation depth of over 10 dB and a fractional bandwidth of 155% within the frequency range of 4.96 GHz to 38.8 GHz, exhibiting tolerance to incident angles and polarization. In the planar state, it achieves strong reflection close to 0 dB over the range of 0-40 GHz, with a fractional bandwidth of 200%. In the folded state, it achieves weak ultra-wideband reflection of less than -10 dB within the frequency range of 4.96-38.8 GHz, with a fractional bandwidth of 155%. Capitalizing on transparent conductive films, high transmittance in the wavelength range of 400-1000 nm is achieved. The authors experimentally verify the ultra-wideband and large-depth reflection modulation performance and analyze it using the theory of multipole decomposition. There are some issues need to be addressed before it can be further considered.

Comment #1

What is the indispensable role of origami in this absorber design? Does the deformation only provide a change in thickness (or the distance to the metallic background), thus improve the absorption bandwidth? If so, this mechanism seems identical to conventional Salisbury screen or Jauman absorbers. The spacer thickness can be easily controlled by filling with foam or dielectric slabs, which has been extensively studied before.

Our Response:

Thanks for providing valuable insights that will enhance our work. The deformation of the origami structure not only induces variations in thickness but also introduces reconfigurable properties and intralayer resonance, distinguishing it significantly from classic absorber designs. We clarify the differences between origami metamaterials and conventional absorber structures from the following three aspects:

(1) **Reconfigurable Properties:** Origami metamaterials naturally possess folding characteristics, allowing them to exhibit reconfigurable electromagnetic properties, particularly suitable for narrow storage space of satellite scenarios. Due to the significant thickness-dependent narrowband absorption in classic absorbers, achieving ultra-wideband reconfigurable reflection is challenging.

(2) **Wideband Absorption and Stability:** Classic absorbers such as Salisbury screens and Jaumann absorbers rely on interlayer resonance for absorption. Hence, their absorption performance is sensitive to thickness. We will take a typical Salisbury absorber as an example, which consists of a lossy layer and a metallic layer sandwich with a 15 mm air gap, as shown in **Fig. R1**. Based on the interference cancellation principle, at 5 GHz, the reflected fields from the lossy layer (E_{LOSSY}) and the metallic ground plane (E_{METAL}) cancel, resulting

in electromagnetic wave absorption. However, at 10 GHz, the change in wavelength disrupts the cancellation performance, leading to wave reflection. Hence, such traditional designs make it hard to achieve continuous ultra-wideband absorption, only allowing for intermittent narrowband absorption. In contrast, origami metamaterials employ intralayer resonance to achieve the 4.96 GHz to 38.8 GHz wideband absorption, different from the interlayer interference. Therefore, our origami structure introduces not only thickness variations but also layer-resonance modes in the folded state. The working mechanism of the proposed origami metamaterial is fully studied through the multipole decomposition theory in the main text.

Fig. R1 The mechanism and absorption performance of a conventional Salisbury absorber.

(3) **Optical Transparency:** The Jaumann screen introduces multilayer layers to achieve wideband absorption. However, this approach leads to a rapid decline in optical transparency. In previous works (Opt. Express 31, 3731-3742, 2023), four layers of transparent conductive films lead to about 40% visible light transmittance, which will decrease the generation efficiency of solar panels. Proposed single-layered origami metamaterials offer over 87% transmittance while maintaining ultra-wideband microwave absorption.

In general, origami metamaterials offer a practical solution for **simultaneously** achieving folding characteristics, ultra-wideband reconfigurable reflection, and high optical transparency. The combination of origami, electromagnetic metamaterials, and flexible electronic materials provides new possibilities for the application of reconfigurable metamaterials in space communication.

Corresponding Revision: We have added more discussions to declare our work innovation.

Revised	Introduction
	These pioneering origami design efforts aim to improve reconfigurability and corresponding core performances without taking optical transparency and process complexity into account³⁵⁻³⁷. However, the practical spaceborne application of origami metamaterials has encountered challenges, especially in achieving optical transparency, ultra-wideband, and large-depth

reflection modulation simultaneously.

Characterization and analysis of practical applicability

While the absorption varies with thickness, the proposed origami metamaterial significantly distinguishes itself from classic Salisbury screens. As an example, the Salisbury screen with origami mechanical support exhibits intermittent narrowband absorption, which depends on interlayer interference, detailed in Supplementary Information S5. As clarified by multipole decomposition, the origami metamaterial in its folded state can effectively achieve ultra-wideband absorption due to continuous mode resonances.

Comment #2

If Miura-ori only provides thickness as mechanical support, with a uniform impedance film placed on top, what are the differences in the absorption performance compared to the device described in the paper?

Our Response:

Fig. S5 | The schematic and absorption performance of the Salisbury screen with origami mechanical support. a The schematic of the Salisbury screen using origami structure as mechanical support with a lossy sheet on the top. **b** The absorption performance of the metamaterial varies with different folded angle β .

Thanks to the reviewer for providing intriguing insights. We conducted simulations to investigate the absorption performance of the proposed origami metamaterial, whose structure is shown in **Fig. S5a**. **Fig. S5b** illustrates the variation in absorption with changing folded angles. The structure's absorption is similar to typical Salisbury absorbers, as the intermittent narrowband absorption varies with thickness. The PET-based origami structure has minimal impact on the absorption performance due to the low dielectric constant ($\epsilon_r = 2.65$) and thin thickness.

For comparison, we present the origami metamaterial and its corresponding absorption in **Fig. R2**. Our folded-state origami metamaterial achieves ultra-wideband absorption without a reflection dip. Meanwhile, the absorption band remains relatively unchanged when the folding angle varies within 60–130°. The comparison

with classic absorbers further highlights the novelty and contribution of our work; this case is presented as Supplementary Information S5.

Fig. R2 | **a** Schematic and **b** the absorption spectral of proposed metamaterial with different folded angle β .

Corresponding Revision:

We made the following main corrections.

Characterization and analysis of practical applicability

While the absorption varies with thickness, the proposed origami metamaterial significantly distinguishes itself from classic Salisbury screens. As an example, the Salisbury screen with origami mechanical support exhibits intermittent narrowband absorption, which depends on interlayer interference, detailed in Supplementary Information S5. As clarified by multipole decomposition, the origami metamaterial in its folded state can effectively achieve ultra-wideband absorption due to continuous mode resonances.

S5: The absorption of the Salisbury screen with origami mechanical support

The Salisbury screen with origami mechanical support is set as a benchmark, whose structure is shown in Fig. S5a. The 0.125 mm-thick PET sheet is used for the origami mechanical support, whose structural parameters are the same as the proposed origami metamaterial. A 0.125 mm-thick PET-ITO sheet with 150Ω/sq surface resistance is placed on the top of the support. By adjusting the folded angle β of the origami mechanical support, it becomes possible to effectively control the thickness of the Salisbury screen, whose absorptions are shown in Fig. S5b. As can be seen, the structure achieves intermittent narrowband absorption, with the absorption peak shiftings as the folded angle changes. The narrowband absorption relies on the interference cancellation between the electric ground and the lossy sheet. As a comparison, the proposed origami metamaterial, in its folded state, achieves ultra-wideband continuous absorption due to the interlayer resonance, as illustrated by multipole decomposition.

Repeated figures and paragraphs from above are not shown.

Revised

Comment #3

Is the metal plate transparent in the wavelength range of 400-1000 nm? In my understanding, the metal background layer is crucial to the absorption performance. Its light transmittance must be discussed if the advantage of optical transparency is claimed.

How does the metal background deform together with the origami layer?

Our Response:

We agree with the reviewer's insights regarding the importance of the metal background layer. In this study, we did not use additional metal backgrounds, as the solar panels can be viewed as metal backgrounds. This concept will be elucidated in the following sections. **Firstly**, we will show the typical solar panel structure, as depicted in **Fig. R3a**. Due to the limited lateral conductivity of the semiconductor layer in solar cells, transparent conductive coatings are necessary as the front electrode to collect the current generated by the solar cell. To reduce the energy loss of solar panels, the surface resistance of solar panels is usually less than $12 \Omega/\text{sq}$. Furthermore, as surface resistance decreases, the shielding effectiveness of the transparent conductive coatings significantly improves, as shown in **Fig. R3b**. When the film's surface resistance falls below $12 \Omega/\text{sq}$, the structure's shielding effectiveness exceeds 20 dB. This result implies that most waves will be reflected when microwave radiation incident the transparent conductive coating. Therefore, solar panels can be considered the metallic ground plane of origami metamaterials, eliminating the need for an additional metallic ground plane. Since there is no additional metal background layer, the problem of how to deform the metal background layer has also been solved.

Fig. R3 a The schematic of a typical solar panel structure. **b** The simulated microwave shielding effectiveness varies with the surface resistance of the transparent conducting coating.

Furthermore, we would like to share the proposed origami metamaterial's robust absorption features, even without the help of the metal background, which is different from classic absorber designs. As shown in **Fig. R4**, the absence of metal ground only results in a 10% absorption decrease compared to the structure with metal ground. Therefore, intralayer resonance of origami metamaterial mainly leads to ultra-wideband

absorption, which is studied through multipole decomposition in the main text. **Hence**, the solar panel can be considered as the metal background for the origami metamaterial. Moreover, the proposed origami metamaterials can work without a metal background, a subject that will be explored in future studies.

Fig. R4 The absorption performance of proposed origami metamaterial with or without ground.

Corresponding Revision: We made the following main corrections.

Before	The periodic boundary is adopted in both x- and y-directions, while an electric boundary is used at the bottom to simulate the solar panel.
Revised	The periodic boundary is adopted in both x- and y-directions to simulate the array effect. Since the solar panel has a high conducting coating that collects the generated current, it can be considered an electric boundary at the bottom of the origami metamaterial.

Comment #4

The Poisson's ratio and Relative area in Figure 5 have both been well studied. What makes them particularly noteworthy or special?

Our Response:

We appreciate the reviewers' question, which further improves the quality of the manuscript. Considering economic factors and the limited storage space on satellites, origami metamaterials should be compressed before rocket launch. In space, origami metamaterials must cover large-scale solar panels to modulate reflection. Hence, we use Poisson's ratio and relative area to show the folding performance. Poisson's ratio represents how the structure changes when stretched in one direction relative to another. Due to the negative Poisson's ratio characteristics of Miura-ori, both the x and y directions will simultaneously expand, which is beneficial for covering large solar panels. Relative area offers a more intuitive parameter, showing the expansion of origami metamaterial. From the compressed state to the planar state, the covering area of origami metamaterial increases from 1 to 162 times.

Corresponding Revision:

In addition to the parameters, we proposed a potential mechanical structure and created a movie demonstrating the scalability of origami metamaterial (Supplementary Movie 2).

Fig. S1 | Conceptual illustration of origami metamaterial folding process on the solar panel of a satellite. a The diagram of the mechanical structure for origami metamaterial deformation. **b-d** The states of deformation for the proposed origami metamaterial in practical space applications.

Reviewer #2:

The authors reported an origami ITO-PET metamaterial with a wideband and large depth reflection modulation function. It was found that when inducing a Miura-fold transition in the metamaterial, a transition from strong reflection of near 0 dB to weak reflection of less than -10 dB could be reached within the frequency range from 4.96 to 38.8 GHz. Moreover, high transmittance of over 87.2% was reached for the metamaterial both in planar and folded states within the wavelength range from visible to near-infrared wavelength. These results were interesting, and the evidences were solid. The manuscript could be accepted after minor Revision.

Comment #1

How about the application scenario of the proposed origami ITO-PET metamaterial in satellite communication. This should be carefully discussed in the manuscript.-

Our Response & Corresponding Revision:

Thanks for your recognition of our work. We have made further efforts in the application scenario of our structure. **Firstly**, we designed a mechanical structure that adjusts the folded angle of the origami metamaterial, as shown in **Fig. S1**. When employed on a solar panel, the folded and planar states of the origami metamaterial can completely cover the solar panel. We have added the mechanical structure and application scenarios of origami metamaterial in Supplementary Information S1. Meanwhile, the folding process of origami metamaterial is demonstrated in Supplementary Movie 2.

Fig. S1 | Conceptual illustration of origami metamaterial folding process on the solar panel of a satellite. **a** The diagram of the mechanical structure for origami metamaterial deformation. **b-d** The states of deformation for the proposed origami metamaterial in practical space applications.

Additionally, we study the impact of the origami metamaterial on the near-field electromagnetic response. In

full-wave simulations, a 10GHz TM-polarized wave is incident at a 45° angle on the solar panel. As shown in **Fig. S2a**, the scattered field effectively propagates along the specular direction. This reflection is effectively suppressed when we load the origami metamaterial structure, as shown in **Fig. S2b**. This phenomenon leads to potential applications:

- (1) **Improve satellite communications quality:** The origami metamaterial effectively suppresses background noise, making communication signals more prominent and improving communication quality.
- (2) **Reduce satellite scattering:** The origami metamaterial can effectively reduce the wideband scattering field, thereby reducing the interference of scattered signals to highly sensitive radio telescopes.
- (3) **Reconfigurable invisibility:** By adjusting the folded angle of origami metamaterial, the satellite can effectively switch between visibility and invisibility state over wideband.

Fig. S2 | The simulated scattering field distribution under 10 GHz planewave oblique incidence. a The solar panel. **b** The solar panel covered by proposed origami metamaterials.

In addition to the two Figures, we also modify our **Introduction** as follows:

However, electromagnetic radiation and scattering from satellites can disrupt radio telescope observations of space, and the demanding space environment presents significant challenges in improving communication quality and managing satellite invisibility^{1,2}. Although incorporating absorption into the mechanical structures has shown promise in reducing scattering fields and improving signal reception sensitivity, their opaqueness and the static nature of electromagnetic signal adjustments limit their applicability to diverse scenarios³⁻⁷.

Besides, we made the following corrections.

Revised	S1: The potential mechanical structures and application scenarios for origami metamaterials As illustrated in Fig. S1, an initial mechanical design with two degrees of freedom is introduced to facilitate the folding of the origami metamaterial above the solar panels. This design enables variations in the folded angle through translation while allowing for rotation, effectively rolling the origami metamaterial into the structure. In this way, the origami metamaterial is in its
---------	---

compressed state before the rocket is launched, reducing the storage space. Once the satellites are deployed in space, the mechanical structure effectively transforms the origami metamaterial from its compressed state to the folded state and planar state covering the solar panel. The detailed deformation process can be found in Supplementary Movies 2.

Repeated figures and paragraphs from above are not shown.

Reviewer #3:

The authors sputtered conductive materials on the surface of the Mirua-ori structure, achieved microwave band modulation by changing the folded angle, and chose transparent material to achieve visible and infrared band transmission. I recommend rejecting the article for publication in Nature Communication for the following reasons.

Comment #1

The idea of sputtering conductive materials on origami structures for wave-absorbing modulation was not innovative enough, although transparent wave-absorbing materials were chosen.

Our Response:

Thanks for the reviewer's comments. To our knowledge, there have been no reports on the combination of transparent conductive films with origami designs, as shown in **Fig. R5**. Existing works using origami structures are often fabricated by 3D printing and PCB process, leading to opaque characteristics. We will discuss the innovativeness of our work from the following three aspects.

Fig.R5 Comparison of the fabricated sample in previous origami metamaterial works and our work.

(1) **Simple structure and low cost:** In many previous works, complex origami structures with intricate metasurface resonators were proposed, significantly raising the complexity of the corresponding deformation mechanical structure and concurrently increasing processing costs. Due to photolithography or 3D printing technology, large-scale production becomes challenging, hindering the practical application of origami metamaterials. In contrast, our proposed metamaterial only requires laser etching creases on an ITO-PET sheet, significantly reducing in processing time and costs. In this work, our structure requires only \$16 per square meter, making it highly practical and applicable.

(2) **Ultra-wideband reconfigurable performance:** In many previous origami structures, they could only operate in a narrow band (PNAS,115,52,13210-13215,2018) (AM,29,27,1700412,2017), making it

challenging to meet practical requirements. In contrast, in our work, the operational bandwidth has been significantly enhanced to 155%, covering the majority of working frequency. In addition, our origami metamaterial exhibits a 10-dB significant modulation depth in reflection over ultra-wideband. The simultaneous presence of ultra-wideband and high modulation depth expands its applicability to a broader range of scenarios.

(3) **Optical transparency required by the practical application:** While there are many available solutions for opacity requirements, controlling microwaves under the requirement of optical transparency remains challenging. In our work, the origami metamaterial achieves an optical transmittance of over 87%, making it feasible for solar panels and optical windows applications.

The difficulty of simultaneously addressing multiple objectives makes achieving the abovementioned advantages challenging. In this work, we integrated flexible electronic materials, electromagnetic metamaterial design, and origami design, enabling reconfigurable satellite scattering and signal enhancement.

Corresponding Revision: The comparison is added to the introduction to state the novelty.

Revised	These pioneering origami design efforts aim to improve reconfigurability and corresponding core performances without taking optical transparency and process complexity into account ³⁵⁻³⁷ . However, the practical spaceborne application of origami metamaterials has encountered challenges, especially in achieving optical transparency, ultra-wideband, and large-depth reflection modulation simultaneously.
---------	--

Comment #2

The origami wave-absorbing structure in the paper has a reflectivity -10 dB bandwidth that does not compare favourably with other wave-absorbing structures at the same thickness, and is sensitive to multi-angle incidence and polarisation of plane waves.

Our Response:

Thanks for the reviewer's comments. The -10dB reflection is a benchmark corresponding to 90% absorption in extensive microwave absorber design. Lower reflections, such as -20dB (99%) or -30dB (99.9%), have limited practical significance in engineering. Therefore, we use -10dB as the benchmark for our metamaterial design.

In contrast to opaque microwave absorbers, transparent absorbers need to balance high optical transparency with wideband absorption. Multilayer lossy film design will significantly reduce transmittance, so research mainly revolves around single-layered transparent absorber designs. **Fig. R6** presents several classic wideband absorbers whose 10dB low-reflection bandwidth improves from 93% to 125%. However, the difficulty of

designing complex metasurface's multi-mode resonance in wideband hinders further bandwidth improvement. Our work offers a more straightforward and robust strategy by introducing origami design into a single-layer film, significantly increasing the absorption bandwidth to 155%. Although the structure's thickness has increased compared to previous works, the structural thickness requirements in our application scenario are flexible. In this work, the proposed metamaterial's absorption bandwidth and transparency are well-balanced without any complex metasurface pattern. What can be expected is that origami metamaterial's performance can be further improved through metasurface pattern design.

Fig. R6 Performance comparison among the transparent absorbers in previous works and proposed origami metamaterials.

Fig. R7 The reflection response of proposed origami metamaterial varies with different oblique angles **a** The reflection spectral of proposed metamaterial under TM wave oblique incidence. **b** The reflection spectral of proposed metamaterial under TE wave oblique incidence.

Our structure exhibits different responses under TE and TM wave normal incidence due to its structural asymmetry. As shown in Fig. R7, under TE wave normal incidence, most of the frequency band still exhibits reflection below -10dB, and only a minor peak appears. The minor peak remains below -7dB, with only a 2.6 GHz-wide reflection over -10dB. Meanwhile, the proposed structure achieves ultra-wideband absorption under TM wave normal incidence. In general, our structure still exhibits good polarization stability.

Moreover, it is noteworthy that symmetric metamaterials commonly exhibit absorption differences under TE

and TM oblique incidence. For instance, in previous works, a symmetrical metasurface absorber shows different absorption under TE and TM wave oblique incidence, as illustrated in **Fig. R8**. Usually, we consider an absorber to have good oblique stability when the absorption remains almost unchanged at oblique angles over 45° . In our structure, under both TE and TM incidence conditions, reflection stays below -10dB (corresponding to over 90% absorption) for most frequencies and at oblique angles up to 60° . Hence, our structure also maintains good oblique stability current performances, which is sufficient to address practical applications.

Fig. R8 The absorption performance of metasurface absorber in reference (OE,28,13,19518-19530,2020). **a** The unit cell of wideband metasurface absorber. The absorption spectral of proposed metamaterial under **b** TE wave and **c** TM wave oblique incidence.

Expectantly, there is still room for further improvement in origami metamaterials. For example, we can enhance polarization stability by designing periodically symmetric origami structures. Meanwhile, considering the introduction of metasurface patterns on Miura-ori may improve angle stability. The platform established by origami structures and metamaterials makes further performance enhancement possible.

Comment #3

Miura-ori is negative Poisson's ratio structure, which cannot completely cover the target metal surface during the folding modulation process.

There is no specific solution for the control of the origami structure; the PET crease is plastic and it may not be easy to maintain the exact folded angle when controlling the whole structure.

Our Response & Corresponding Revision:

We agree with the reviewer's opinion regarding the change in coverage area during the folding process. The Miura-ori is a negative Poisson's ratio structure, and its covering area significantly changes when transitioning between the folded and planar states. To address this problem, we propose a mechanical structure for the

transition between the compressed, folded, and planar states. This design ensures that both folded and planar states can cover the entire solar panel surface. We have added the mechanical structure and application scenarios of origami metamaterial in Supplementary Information S1 and corresponding Supplementary Movie 2.

Fig. S1 | Conceptual illustration of origami metamaterial folding process on the solar panel of a satellite. **a** The diagram of the mechanical structure for origami metamaterial deformation. **b-d** The states of deformation for the proposed origami metamaterial in practical space applications.

Revised	S1: The potential mechanical structures and application scenarios for origami metamaterials As illustrated in Fig. S1, an initial mechanical design with two degrees of freedom is introduced to facilitate the folding of the origami metamaterial above the solar panels. This design enables variations in the folded angle through translation while allowing for rotation, effectively rolling the origami metamaterial into the structure. In this way, the origami metamaterial is in its compressed state before the rocket is launched, reducing the storage space. Once the satellites are deployed in space, the mechanical structure effectively transforms the origami metamaterial from its compressed state to the folded state and planar state covering the solar panel. The detailed deformation process can be found in Supplementary Movies 2. Repeated figures and paragraphs from above are not shown.
---------	---

Based on the proposed mechanical structure, the origami metamaterials can be arbitrarily controlled without stringent requirements. On the one hand, the robust absorption in the folded state from 60 to 130 degrees provides the mechanical structure with sufficient tolerance. On the other hand, according to Supplementary Movie 1, the fabricated origami metamaterial can effectively be deformed by hand without too much force, further reducing the power requirements on the mechanical structure. **In summary, the proposed origami**

metamaterial can effectively switch states by the mechanical structure while covering the whole solar panel. The mechanical structure proposed above is a preliminary design. It is expected that adaptive control of origami metamaterials may be achieved through liquid crystal elastomers or pneumatic structures in the future.

Comment #4

The design of this wave-absorbing origami structure does not take into account its reliability, the difference in temperature between day and night in space applications can be up to 300 °C, which PET may not be able to cope with, as well as the effects of high-energy rays from space.

Our Response& Corresponding Revision:

Thanks for the reviewer's question. Indeed, the response of materials to extreme temperature variations in space environments is a critical concern. In recent years, flexible sheet materials like ETFT, PET, and PI have widely been used to fabricate flexible solar panels. And some helpful attempts have been made to use flexible solar panels in actual satellites. For example, the SolarFlex flight model flew on ASTRA-1Q in November 2021^[1]. Therefore, the preliminary validation of the application of flexible transparent materials like PET in space has been achieved.

The reason that PET can be used in space is its ability to maintain good physical properties over a wide temperature range^[2]. PET material can withstand high temperatures of up to 150°C (302°F) and low temperatures as low as -70°C (-94°F), with minimal impact on its mechanical performance. Therefore, PET can effectively withstand significant temperature variations in space. As a comparison, other tunable materials like liquid crystals (-20°C~70°C) and vanadium oxide (>68°C) have much narrower operating temperature ranges and poorer mechanical properties.

We attempted to find the PET's high-energy ray resistance characteristics, but did not find direct reports. Nevertheless, we discovered strategies for conferring radiation resistance to PP (polypropylene) material and other plastic materials, as documented in publications from the last century^[3,4]. We believe that research on the radiation resistance of PET is fundamental for researchers with a background in chemical engineering, or the industry may have already addressed this aspect.

Revised	Due to the robust mechanical performance under large-temperature differences, the origami substrate uses 0.125 mm-thick PET and is covered by a 20 nm-thick ITO film with a surface resistance of 150 Ω/sq.
---------	---

Reference:

[1] Flexible Solar Panels Application: <https://www.thalesaleniaspace.com/en/news/solarflex-solar-arrays->

set-fly-space-inspire-satellites

[2] PET Characteristics: [https://www.tekra.com/sites/default/files/downloads/Tek Tip - PET Temperature Range.pdf](https://www.tekra.com/sites/default/files/downloads/Tek_Tip_-_PET_Temperature_Range.pdf)

[3] Wündrich, Konrad. "A review of radiation resistance for plastic and elastomeric materials." *Radiation Physics and Chemistry* (1977) 24.5-6 (1984): 503-510.

[4] Rolando, Richard J. "Radiation Resistant Polypropylene-New Developments." *Journal of Plastic Film & Sheeting* 9.4 (1993): 326-333.

Reviewer #4:

The work leverages origami structure to create a foldable metamaterial that enables electromagnetic ultra-wideband and deep reflection modulation. The optimized material can achieve a modulation depth exceeding 10 dB within the microwave telecommunication range, showcasing its potential for cost-effective and lightweight application in satellite frequency bands. The findings and methodology are sound.

Comment #1

Given that various origami structures have previously been employed in the creation of reconfigurable electromagnetic devices with diverse functions, particularly in microwave range [Adv. Mater.2017, 29, 1700412; J. Phys. D: Appl. Phys. 54, 165111 (2021); Opt. Lett. 46, 1349 (2021); IEEE Trans. Antennas Propagat. 70, 4558–4568 (2022); Sci. Rep. 12, 13449 (2022)]. The work shows promise in practical application, but it needs to differentiate itself from prior work in the field. Therefore, I am uncertain if the significance of this work justifies its publication in Nature Communications. The following comments are listed for the authors' reference.

Our Response:

Thanks for the reviewer's question, and we are glad to clarify the innovation in our work. In **Fig. R5**, we have shown all the origami structures mentioned by the reviewer.

- Among previous works using origami design, all samples are fabricated using conventional opaque materials, which limited their applications in transparent windows or solar panels.
- Previous works use complex origami and metasurface structure design, leading to complex manufacturing processes, longer production times, and higher costs.
- Complicated theoretical models are introduced in previous works to reveal the working mechanism of the origami structures, which are difficult to use and have certain limitations.

Fig. R5 The Comparison of the fabricated sample in previous works and our work

In our work, we solve these three issues mentioned above. **Firstly**, we employed transparent conductive film to origami metamaterial, making it possible for optical transparency and EM wave modulation applications. **Secondly**, our origami metamaterial uses a single-layer ITO-PET film while avoiding the time-consuming and expensive processes. Unlike slow 3D printing, the proposed process can be directly applied to high-speed and large-scale production. Simple structure, processes, and cost-effective materials will significantly reduce the manufacturing cost of the origami metamaterial. The manufacturing cost mentioned in the article is only \$16 per square meter. **Finally**, we introduced the multipole decomposition theory, providing a universal theory tool for future origami metamaterials analysis and design.

In addition to the design, we manufactured large-sized samples and conducted comprehensive validation. In this revision, we further clarified the practical application scenarios of origami metamaterials in space, providing valuable insights into the actual application of metamaterials.

Corresponding Revision:

Introduction

These pioneering origami design efforts aim to improve reconfigurability and corresponding core performances **without taking optical transparency and process complexity into account**³⁵⁻³⁷. However, the practical spaceborne application of origami metamaterials has encountered challenges, **especially in achieving optical transparency, ultra-wideband, and large-depth reflection modulation simultaneously**.

S1: The potential mechanical structures and application scenarios for origami metamaterials

Fig. S1 | Conceptual illustration of origami metamaterial folding process on the solar panel of a satellite.
a The diagram of the mechanical structure for origami metamaterial deformation. **b-d** The states of deformation for the proposed origami metamaterial in practical space applications.

Revised

Comment #2

The authors referred to the optimal beta angle for achieving wideband weak or wideband strong reflection through the optimization process on Page 4. Could the authors provide more details about the specific optimization technique employed in this process?

Our Response:

We are willing to share the optimization details. The folded angle β for wideband absorption is achieved through the optimization process shown in **Fig. R9**. By using a genetic algorithm, we can find the folded angle β that maximizes the bandwidth of reflections below -10dB within the range of 0–180°. Hence, the origami metamaterial with a folded angle of 95° is confirmed as the folded state. Similarly, the reflective state is confirmed by maximizing the bandwidth of reflections over -1dB. Notably, the reflection of the origami metamaterial is simulated by the commercial software in the optimization process.

In the genetic algorithm, binary tournament selection is utilized to remove the worst individual during the evolution stage. Analog binary crossover is implemented with a probability of 70% to combine binary code from different individuals, promoting diversity and exploring new solutions. Subsequently, the polynomial mutation is applied with a probability of 1/25 to introduce random changes in the binary code, preventing premature convergence and preserving the elite strategy. Since our work does not primarily focus on the optimization of the origami metamaterial, we will provide a brief overview of the optimization settings and cite our previous optimization work.

Fig. R9 The optimization process of designing origami metamaterial.

Corresponding Revision:

Before	Through the optimization process, the folded state is determined to be $\beta = 95^\circ$ for wideband weak reflection, while the planar state corresponded to $\beta = 180^\circ$ for wideband strong reflection.
Revised	Using the genetic algorithm, the reflection bandwidth below -10 dB is maximized for wideband absorption while also maximizing the bandwidth of reflections above -1dB to achieve wideband reflection ³⁸ . Hence, the folded state and planar state of origami metamaterial are determined to be $\beta = 95^\circ$ and 180° , respectively.

Comment #3

This work emphasizes its application in microwave telecommunication. Considering that the metamaterial's 'floor area' is reduced when folded, how can we ensure a fair and accurate evaluation of the material's performance across different folded angles, for example, energy output?

Our Response & Corresponding Revision:

Fig. S1 | Conceptual illustration of origami metamaterial folding process on the solar panel of a satellite. a The diagram of the mechanical structure for origami metamaterial deformation. **b-d** The states of deformation for the proposed origami metamaterial in practical space applications.

Thanks for the reviewer's suggestion. The origami metamaterial's area change affects the overall energy reflection when the metamaterial cannot cover the whole solar panel. To address this, we have proposed a mechanical structure that allows full solar panel coverage with the origami metamaterial in both the folded and planar states. Meanwhile, the part of origami metamaterial beyond the solar panel reflects electromagnetic waves to a minimal extent. Hence, the actual reflection on the solar panel will be very close to the results shown in the paper. Further details of the mechanical structure can be found in Supplementary Information S1.

Comment #4

Referring to both Fig. 2b and Fig. 2c, it's evident that the spectral features in the microwave range and the near-infrared range are notably distinct. I'm curious about how the reflectance spectrum might differ if matrix theory were utilized instead of full-wave simulations to investigate these features in the microwave range.

Our Response:

Thanks for the interesting question. We conducted matrix calculations and simulations on a flat $150\Omega/\text{sq}$ film structure in the air, where A and T represent the absorptance and transmittance of the structure, respectively. As shown in **Fig. R10**, the results showed a perfect match between simulation and matrix theory. However, the two methods produced different results when we used the approximation method to calculate the microwave transmittance and reflectance in the folded state, as we used in optical wavelengths. The difference is because the dimensions of the origami metamaterials are similar to the incidence wavelength; the generating mode resonance cannot be solved through matrix theory.

Fig. R10 The simulated and calculated reflection (R) and transmittance (T) of proposed origami metamaterial in its **a** planar and **b** folded state.

To sum up, there are two reasons that matrix theory is less applied in microwave designs:

- **Limitations of matrix theory in solving structures:** As seen in the Comparison, matrix theory effectively solves multilayer structures but struggles to analyze resonances in subwavelength structures like metamaterials.
- **Difficulty in extracting material properties:** In microwave matrix calculations, the material properties of ITO are defined through permittivity, permeability, and conductivity. However, obtaining the parameters of films in practical microwave measurements is challenging due to thin film thickness and non-uniform distribution.

Comment #5

Given that both the planar and folded states demonstrate high transmittance in the visible and near-infrared range (as shown in Fig. 2c), what specific advantages does the origami metamaterial offer in this context?

Our Response:

Since solar cells' spectral response wavelength range is around 400–1200nm, the design of structures covering the solar panel needs to satisfy transparency in this wavelength range to avoid affecting power generation. In this work, practical solar panel experiments have verified the proposed origami metamaterial's transparency property, as shown in Supporting Information S3. This demonstrates that origami metamaterials do not impact the power generation of satellite solar panels in space while effectively achieving ultra-wideband and large-depth reflection modulation.

Comment #6

The color variation in the colorbar, ranging from -3 to -20 dB in Figure 4c, appears limited. Enhancing the colorbar settings could potentially provide a richer representation of the data. Additionally, in Figure 4d, there seem to be abrupt transitions in the colormap. It may be beneficial to consider refining the plot or providing additional explanation for these sharp changes.

Our Response:

Throughout all the figures, we aimed to guide the reader with colors, specifically using orange to represent absorption and green to represent reflection. This was the primary reason for choosing this colormap for Fig.4c and 4d.

Fig. R11 **a** The absorption spectral of origami metamaterial with different β . **b,c** The reflection spectral of origami metamaterial with different β and colormap.

In the subsequent comparison, we will demonstrate the fairness of our colormap setting in **Fig. 4c**. First, we used a color map with absorption as the benchmark, where the white is at an absorption of 0.7, as shown in **Fig. R11a**. It can be observed that the structure exhibits wideband absorption characteristics when β is in the range of 30–130 degrees. Through the Comparison between **Fig. R11a** and **b**, we can find that the reflection spectra closely match the color distribution of the absorption map. This shows that the colormap of the reflection spectra is set fairly, showing wideband absorption performance. While the white color is set at -

10dB, and the color distribution for reflection is notably different. However, we can still find the robust wideband absorption of proposed origami metamaterial through **Fig. R11c**. The main reason the colormap appears unusual is that reflection is represented using reflection in dB scale (10dB for 90% absorption, 20dB for 99% absorption). In fact, the difference in energy between -10dB and -20dB absorption is not substantial; hence, they can share a similar color. The use of a 3dB (50% absorption) dividing line in the figures conveniently separates absorption (>50% absorption) and reflection (<50% absorption) fairly.

In absorber design, we usually consider bandwidth exceeding 100% as wideband design. Hence, the benchmark is set at 100% fractional bandwidth, denoted by the white color in **Fig. 4d**. In fact, the existing colorbar settings can distinguish the difference between broadband and narrowband. We hope reviewers will acknowledge our efforts to facilitate quick comprehension for general readers.

Comment #7

While the reported structure can be expanded in one direction, it would be interesting to know if the folded structure, which can be expanded along both the x- and y-directions, exhibits similar properties. Have these properties been investigated or considered?

Our Response:

Fig. R12 The period of the unit cell along the x- and y-axis verifies with the folded angle β .

The simultaneous expansion along both the x and y directions is an interesting property of Miura-ori. To visually demonstrate this property, we calculated the variation in the unit cell's x and y directions as a function of folded angle β in **Fig. R12**. It can be observed that the period change along the x-direction is quite significant, while the period change along the y-direction is comparatively smaller. This result is consistent with the deformation of origami metamaterials on solar panels. Of course, by adjusting the structural parameters of Miura-ori, the period length of units at different folded angles can be further adjusted. However, the expansion in the structure in the y direction will always be more significant than in the x direction, which is limited by

the intrinsic properties of Miura-ori. Meanwhile, the change in the unit cell size will also influence its electromagnetic absorption response, which will be explored in our future work.

Comment #8

It would be helpful to include the full names of the labels 'A,' 'R,' and 'T' in the caption of Figure 2 or within the main text for clarity and better understanding.

Our Response & Corresponding Revision: We have added this part to the figure caption.

Revised

Fig. 2 | c,f The proposed metamaterial's calculated and measured absorptance (A), reflectance (R), and transmittance (T) in visible to near-infrared light.

Comment #9

Since Equation 1 is featured in the main text, it would be beneficial to provide the definition of each component within the main text, even though they are already given in the supplementary materials. This would enhance the accessibility of the information for readers.

Our Response & Corresponding Revision: We have added definitions in the main text.

Revised

By solving the current density of the origami structure as input through commercial software, multipole decomposition theory is used to classify the currents and determine their contributions to the total power, as illustrated in Equation (1).

(1)

where P_{ED} and P_{MD} represent the power of electric and magnetic dipole, P_{EQ} and P_{MQ} represent the power of electric and magnetic quadrupole, P_{EO} and P_{MO} represent electric and magnetic octupole, respectively. Further detailed derivation is provided in Supplementary Information S4.

Comment #10

Consider revising certain terms or sentences to enhance accessibility for general readers. For example, "non-pol", "Visible-near-infrared light measurement"

Our Response & Corresponding Revision: Thanks for your suggestion; we are attempting to present those

terms in a more accessible and understandable manner.

Before	Fig. 2c presents the non-pol optical spectrums of metamaterial in its planar and folded states. Microwave measurement Visible–near–infrared light measurement
Revised	Fig. 2c presents the unpolarized optical spectrums of metamaterial in its planar and folded states, which can be regarded as the averaged spectra of TE and TM light. Microwave reflection measurement Visible–near-infrared reflectance and transmittance measurement

Comment #11

Gramme errors. For example, "In the range of folded angle beta is between 80 and 110°", "Comparison of different approach to dynamic modulation metamaterial"

Our Response & Corresponding Revision: Thanks for the reviewer's comments. We have addressed the grammatical errors as outlined below.

Before	 1. To validate the design, we fabricate the origami metamaterial sample using 400 mm × 500mm ITO-PET sheets with a surface resistance of 153.5 Ω/sq resulting in an 8×10 array configuration. 2. Without shaded, the solar panel achieve an energy supply of 9.90 mW under simulated sunlight irradiation. 3. In the range of folded angle β is between 80 and 110°, the structure's thickness remains relatively unchanged, providing favorable conditions for wideband absorption. 4. Comparison of different approach to dynamic modulation metamaterial.
Revised	 1. To validate the design, we fabricate the origami metamaterial sample using 400 mm × 500mm ITO-PET sheets with a surface resistance of 153.5 Ω/sq, resulting in an 8×10 array configuration. 2. The solar panel achieves an energy supply of 9.90 mW without shade under simulated sunlight irradiation. 3. When the folded angle β is between 80 and 110°, the structure's thickness remains relatively unchanged, providing favorable conditions for wideband absorption. 4. Comparison of different approaches to dynamic modulation in metamaterials

REVIEWERS' COMMENTS

Reviewer #1 (Remarks to the Author):

Good answers are provided to all comments, and revisions have been made to address the corresponding technical details. We recommend acceptance if the editor deems the novelty sufficient to meet requirements of nature communications.

Reviewer #2 (Remarks to the Author):

The authors have answered all my questions. The revised manuscript could be accepted.

Reviewer #4 (Remarks to the Author):

The authors have addressed my concerns and I have no additional questions. I'd like to give my recommendation for the publication.